# Efficacy of smartphone-based virtual reality relaxation in providing comfort to patients with cancer undergoing chemotherapy in oncology outpatient setting in Indonesia: protocol for a randomised controlled trial

Made Satya Nugraha Gautama,[1,2] Haryani Haryani,[3] Tsai-Wei Huang [1,4,5,6]

For numbered affiliations see end of article.

**Correspondence to**
Professor Tsai-Wei Huang;
tsaiwei@tmu.edu.tw

## ABSTRACT

**Introduction** Patients with cancer undergoing chemotherapy experience various physical and psychological problems and discomfort. Virtual reality (VR) can be used in technology-based non-pharmacological therapy that can serve as a potential distractor in the symptom management of patients with cancer undergoing chemotherapy. We propose a smartphone-based virtual reality relaxation (S-VR) technique as a complementary modality to provide comfort to patients with cancer, and we will evaluate its effect on patients with cancer undergoing chemotherapy.

**Methods and analysis** We will recruit 80 patients from the One Day Chemotherapy 'Tulip' Center of Dr. Sardjito General Hospital, Yogyakarta, Indonesia. This will be a two-arm parallel randomised controlled trial, with a 1:1 allocation and the primary outcome assessor blinded. This study will be divided into two groups: (1) an intervention group, with participants receiving 360° panoramic video content and music relaxation intervention through a VR device (head-mounted display) placed on their head during chemotherapy for ±10 min plus standard care and (2) a control group, with participants receiving guided imagery relaxation therapy in the form of a leaflet plus standard care. We will measure the outcomes after one chemotherapy cycle for each participant. The primary outcome is the effectiveness of the S-VR in improving the comfort of patients. The secondary outcome is the effect of the S-VR on the patients' symptom management self-efficacy, pain, anxiety, blood pressure (systolic blood pressure and diastolic blood pressure) and pulse rate.

**Ethics and dissemination** This study was approved by the Medical and Health Research Ethics Committee of the Faculty of Medicine, Public Health and Nursing of Universitas Gadjah Mada—Dr. Sardjito General Hospital, Yogyakarta, Institutional Review Board (approval number: KE/FK/0301/EC/2023). Written informed consent will be obtained from all participants who enrol in the study. Dissemination will be conducted through peer-reviewed publications and conference presentations.

**Trial registration number** NCT05756465.

## STRENGTHS AND LIMITATIONS OF THIS STUDY

⇒ In addition to patient-reported outcomes, the present randomised controlled trial will assess patients' bio-parameters, such as blood pressure and heart rate, as well as cybersickness symptom evaluation due to the virtual reality (VR) device.
⇒ We will increase the sample size and make it larger than minimum sample size to increase statistical power.
⇒ The principal objective of this study was to enhance patient comfort during the brief duration of chemotherapy within the chemotherapy room.
⇒ We will not assess the patients' pre-existing psychological conditions, which may affect their response to VR intervention.
⇒ This study conducts in a single institution, which may limit the generalisability of the findings to other settings and populations.

## INTRODUCTION

The cancer incidence in Indonesia (136.2/100 000 population) is the 8th highest in Southeast Asia and 23rd highest in Asia.[1] According to Basic Health Surveillance (Riskesdas),[2] the prevalence of cancer in Indonesia increased from 1.4 in 2013 to 1.49 in 2018. Of the total of 396 914 cancer cases in Indonesia in 2020, the top three most common cancer types were breast cancer (65 858 cases; 16.6%), cervical cancer (36 633 cases; 9.2%) and lung cancer (21 392 cases; 8.8%).[3] The region with the highest prevalence of cancer is the Special Region of Yogyakarta (DIY) Province (4.86/1000 population), and 23% of the patients with cancer in this region receive chemotherapy, which is the second-most-common cancer management strategy after surgery.[2]

Patients undergoing chemotherapy experience various adverse effects.[4] Moreover, they may experience physical and psychological discomfort caused by the treatment drugs, invasive procedures and chemotherapy environments. In general, 86% of patients with cancer experience pain, nausea, vomiting, mucositis, diarrhoea and alopecia.[5 6] Other side effects include psychological problems, such as anxiety, feelings of hopelessness, depression, anger, fear, vulnerability and a sense of loss of control.[7] Management of the side effects of chemotherapy by using drugs is not fully effective and may cause other side effects, which can cause further patient discomfort.[8 9]

Comfort is defined as a lack of pain, distress, worry and uneasiness, and it is one of the most crucial factors in holistic nursing care.[10 11] Focusing on patients' biophysiological aspects (eg, symptom management) as well as psychospiritual, environmental and sociocultural aspects related to healing is crucial for maintaining their wellbeing.[12] Kolcaba explained that in addition to providing basic care to patients, nurses must provide special and unique nursing care to each patient to increase patient comfort.[13 14] Patients with cancer require physical, social and spiritual support and a comfortable environment to have a sense of life meaning.[12 15 16] Meeting the comfort needs of patients with cancer positively affects the patients' satisfaction with the services they receive, their treatment adherence and their overall quality of life.[11 17]

Many non-pharmacological or alternative approaches are used in the management of chemotherapy side effects.[18] Some studies have reported that non-pharmacological therapies, such as music therapy, acupuncture, massage, guided imagery, hypnosis, acupressure and virtual reality (VR) therapy, can alleviate physical and psychological symptoms in patients with cancer.[19 20] However, most non-pharmacological therapies involve the use of conventional methods; few have incorporated technologies, such as VR, into alternative therapy.[21] The use of VR in both cancer research and therapy remains limited.[22]

VR therapy is gaining popularity in complementary medicine due to its ability to overcome certain limitations of traditional therapy.[23 24] First, VR offers an immersive and interactive environment that fosters a sense of presence and transports individuals to a different reality, potentially reducing anxiety and pain perception.[25] Research has shown that this immersive experience can create a therapeutic effect. Second, VR therapy provides a customisable and controlled environment that can be tailored to meet the specific needs of individual patients.[26] This personalised approach allows for a more targeted and effective treatment experience. By adapting the virtual scenarios and activities to the patient's preferences and requirements, VR therapy offers a unique and individualised intervention. Although VR has been partially applied in cancer care, especially to reduce patient anxiety and pain during invasive procedures.[21 27]

Although scientific evidence of the effectiveness of using VR in the treatment of cancer continues to grow, findings regarding the use of smartphone-integrated VR remain limited.[28 29] Therefore, in the present study, we proposed an innovative strategy for the management of discomfort in patients undergoing chemotherapy by using smartphone-based virtual reality relaxation (S-VR) with content including natural panoramic views and three-dimensional relaxation music. It is anticipated that this alternative therapy will help alleviate symptoms in chemotherapy patients by reducing anxiety and diverting attention from pain. However, further research is needed to explore the full potential and efficacy of smartphone-based VR therapy in managing chemotherapy-related symptoms.

The primary objective of this study is to determine the efficacy of S-VR in increasing comfort in patients with cancer undergoing chemotherapy. The secondary objectives of this study are to assess the effects of S-VR on the symptom management self-efficacy, anxiety, pain and vital signs (pulse and blood pressure) of patients with cancer undergoing chemotherapy. We hypothesise that relative to the control group, the patient group that receives the S-VR intervention will experience greater improvements in their comfort levels.

## METHODS AND ANALYSIS
### Design
This study is a two-arm parallel randomised controlled trial with a 1:1 allocation. The study will divide the patients into two groups: the intervention group, in which the patients will receive S-VR therapy through a VR device (head-mounted display) mounted on the patients' heads during chemotherapy plus standard care (daily routine care in administration of chemotherapy drugs), and the control group, in which the patients will receive standard care and guided imagery therapy with leaflets containing instructions for techniques for relaxing through image visualisation. The group allocation will be implemented using the envelope concealment method, and the outcome assessor will be blinded to the group allocation during data collection. The protocol has been designed in accordance with the Standard Protocol Items: Recommendations for Interventional Trials and its checklist.[30 31] Figure 1 presents the modified Consolidated Standards of Reporting Trials[32] flow diagram.[32]

### Setting
The DIY Province will be the study location because it has the highest prevalence of cancer in Indonesia. The national prevalence of cancer is 1.79/1000 population, whereas the prevalence in the DIY Province is 4.86/1000 population, with 23% of these patients undergoing chemotherapy treatment.[2] This study will be conducted at the One Day Chemotherapy Polyclinic 'Tulip' Integrated Cancer Unit of Dr. Sardjito General Hospital, Yogyakarta, Indonesia. According to the annual report of the service performance of the Integrated Cancer Unit of Dr. Sardjito General Hospital, the number of visits by patients

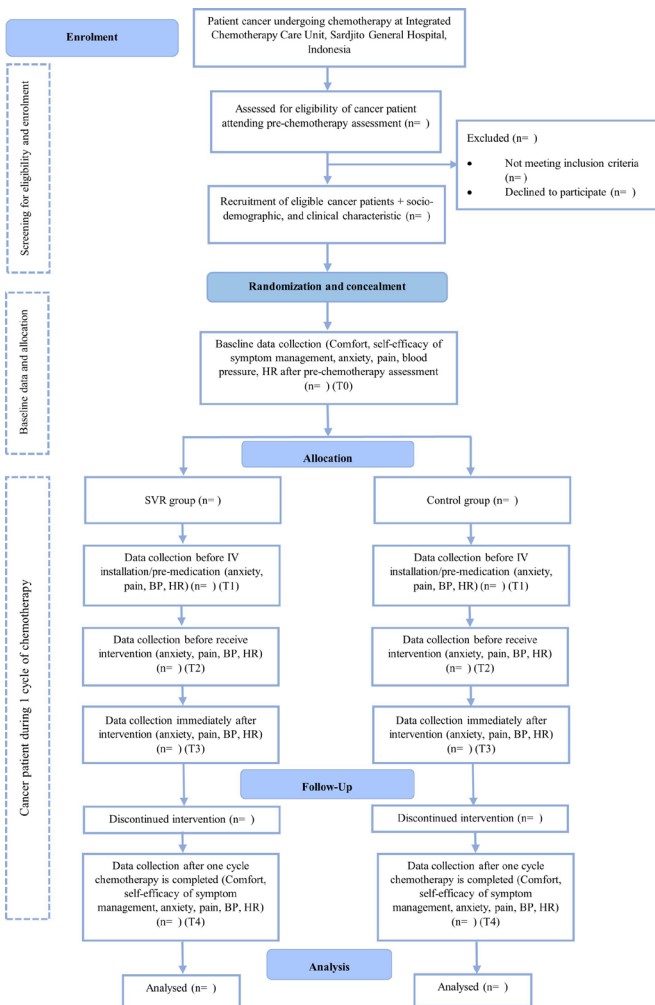

**Figure 1** Modified Consolidated Standards of Reporting Trials flow diagram.

with cancer was 100 614 people in 2019, an increase of 19.8% from the previous year.

### Patient and public involvement

The protocol has been designed in close collaboration between the School of Nursing, Taipei Medical University, and the Master of Nursing Programme, Universitas Gadjah Mada, with strong support from Dr. Sardjito General Hospital, Yogyakarta, Indonesia. We work with patients with cancer undergoing chemotherapy under the Integrated Cancer Unit at Dr. Sardjito General Hospital, Yogyakarta, Indonesia. A patient with cancer undergoing chemotherapy was enrolled in this intervention programme and completed the study evaluation. Valuable input, preference and evaluation on the part of both patients and hospital institutions have been invaluable in the development of this intervention; thus, we hope that the findings from this randomized controlled trial will be disseminated to local, regional and national policymakers as well as an international scientific audience.

### Population and sample

The target population is patients with cancer who will undergo chemotherapy at the 'Tulip' Integrated Cancer

Unit of Dr. Sardjito General Hospital. The inclusion criteria for this study are as follows:
1. Being a patient with cancer receiving chemotherapy in all cycles.
2. Being aged ≥18 years.
3. Having a performance status (Eastern Cooperative Oncology Group, ECOG Score) of 0–2.
4. Being able to understand and sign the informed consent form.

Respondents who meet the following exclusion criteria will be excluded from the study:
1. Being a patient with cancer with a history of skull structure or cervical spine abnormalities that may complicate the use of VR devices.
2. Being a patient with cancer with cognitive, visual and hearing impairment; or having a chemo port.
3. Being a patient with a history of epilepsy, seizures, vertigo or visually induced motion sickness.

### Sample size calculation

The sample size was calculated using free-to use software, namely G*Power for Windows V.3.1.9.6.[33] The effect size is 0.81, type 1 error ($\alpha$)=5% (two-tailed), power (1−$\beta$) 90% and the allocation ratio between the groups is 1:1. The effect size was set with reference to previous studies with similar outcomes.[9] According to the G*Power calculation, the study sample size should be 34 per group. With consideration of an expected dropout rate of 15%, the sample size was increased to 40 per group.

The participant dropout criteria are as follows:
1. The patient being unwilling to continue the sequence of the study process (lost to follow-up).
2. The patients experiencing drug extravasation incidents or severe side effects during chemotherapy.

The consecutive sampling technique will be applied. This non-probability sampling method involves recruiting all individuals from the accessible population during a given time interval or for a specific sample size. Patients with cancer who have scheduled chemotherapy and meet the requirements of the data collection time will be asked to participate in the study.

### Participant characteristics and study outcomes
#### Participant characteristics

A questionnaire will be used to collect patient demographic data, including age, sex, education level, marital status, occupation, income and assistance during chemotherapy, and clinical characteristics, including the type of cancer, body surface area (height and weight), cancer stage, duration of diagnosis, chemotherapy cycle and ECOG Score.

#### Outcomes

The primary and secondary outcomes are summarised in table 1.

Comfort of patients with cancer is measured using the Shortened General Comfort Questionnaire (SGCQ) instrument by Katharine Kolcaba in 2006 which has been

**Table 1** Study outcomes

| Outcome | Measurement tool | Score | Interpretation |
|---|---|---|---|
| **Primary** | | | |
| Comfort | The Shortened General Comfort Questionnaire, Indonesian version[41] | Likert Scale—1 (strongly disagree) to 6 (strongly agree) | Higher scores indicate higher levels of comfort. |
| **Secondary** | | | |
| Symptom management self-efficacy | The Self-Efficacy of Cancer Symptom Self-Management Scale[42] | Likert Scale—1 (not at all sure) to 10 (very sure) | Higher scores indicate higher levels of self-efficacy. |
| Anxiety | The Visual Analogue Scale for Anxiety[43 44] | A horizontal line with a scale of 100 mm | Higher scores indicate greater anxiety. |
| Pain | The Numerical Pain Rating Scale[45] | Numerical scale—0 to 10 (11-point scale) | Higher scores indicate greater pain. |
| Pulse rate | Digital sphygmomanometer | Beats per minute (bpm) | An adult's regular resting heart rate ranges from 60 bpm to 100 bpm. |
| Blood pressure | Digital sphygmomanometer | Millimetre of mercury (mm Hg) | Systolic pressure above 90 mm Hg and lower than 120 mm Hg and diastolic pressure between 60 mm Hg and 80 mm Hg |

tested for validity and reliability in Indonesian version by Artanti *et al*.[34] The validity test reported that Content Validity Index (CVI) of Item (I-CVI) and Scale (S-CVI) Score were both 1 (valid). Cronbach's α score showed 0.769, meaning that the SGCQ Indonesian version is reliable. The SGCQ instrument describes individual comfort by assigning a Likert Score to 28 items. Likert Scores range from 1 (strongly disagree) to 6 (strongly agree). The highest total score is 168, and the lowest is 28.

Self-efficacy is measured by using the Self-Efficacy of Cancer Symptom Self-Management Scale (SE-CSSMS),[35] which consists of six question items with a value range of 1 (not at all sure) to 10 (very sure). SE-CSSMS has been tested for validity and reliability on patients with cancer in Indonesia.[35] The CVI is 0.94. The r value is 0.467, which is greater than the r-table value of 0.12, so these results state that the six items in the symptom management self-efficacy instrument are valid. The reliability test results showed that the Cronbach α value is 0.90 (>0.70), so the measuring instrument is reliable.

Anxiety is measured using the Visual Analogue Scale for Anxiety (VAS-A) in the form of a horizontal line with a scale of 100 mm or 10 cm, with the left by the words 'not at all anxious' and on the right by 'extremely anxious'.[36 37] VAS-A has a sensitivity of 76.8%, specifications of 64.9% and internal reliability consistency (Cronbach's α ≥0.7).[36 37]

Pain is measured using the Numerical Rating Scale (NRS). The NRS consists of numerical scale; from 0 to 100 (101-point scale), 0 to 10 (11-point scale) or 1 to 10 (10-point scale). This study will use a 0–10 NRS due to it provides more sufficient sensitivity to measure pain in most patients with cancer.[38] The NRS has been declared valid and reliable for subjective measurement of cancer

pain, both due to adverse effects of cytotoxic agents, cancer treatment procedures (surgery, radiotherapy and chemotherapy) and the malignancy of the cancer itself.[39]

Pulse rate and blood pressure (systolic and diastolic pressure) are measured using an automatic digital sphygmomanometer. According to American Heart Association, the regular resting heart rate for adults is 60–100 beats per minute, and systolic blood pressure is above 90 mm Hg and less than 120 mm Hg, with a diastolic pressure between 60 mm Hg and less than 80 mm Hg.[40]

Comfort and self-efficacy for symptom management will be measured at two time points: at baseline (before chemotherapy premedication) and post-test (after the chemotherapy session is completed). Anxiety, pain and vital signs (pulse rate and blood pressure) will be measured at five time points: thrice at pretest (baseline data collection, before the installation of intravenous access and before administration of VR intervention) and twice at post-test (immediately after intervention and after the chemotherapy session is completed). The timeline of the data collection and measurement is presented in table 2.

### Randomisation and concealment

The researchers will screen to identify patients who are eligible to participate in the study. The patients' medical records will be evaluated to verify their baseline information (eg, anthropometric data, laboratory data, patient identification data and laboratory results). An allocation sequence will be generated through simple randomisation of the study participants by using a formula on the Microsoft Excel program (the RANDBETWEEN function), and the allocation sequence will be concealed in sealed envelopes. Thereafter, the participants in the intervention and

**Table 2** Timeline of data collection and measurements

| Assessment/ measurement instrument | Study period | | | | |
|---|---|---|---|---|---|
| | T0 | T1 | T2 | T3 | T4 |
| Demographic variables | X | | | | |
| Clinical characteristics | X | | | | |
| SGCQ | X | | | | X |
| SE-CSSMS | X | | | | X |
| VAS-A | X | X | X | X | X |
| NPRS | X | X | X | X | X |
| Blood pressure | X | X | X | X | X |
| Pulse rate | X | X | X | X | X |

T0: baseline data collection (after prechemotherapy assessment and obtaining signed informed consent); T1: before intravenous installation/premedication; T2: before the start of the intervention; T3: immediately after the intervention; T4: after completion of one chemotherapy cycle.
NPRS, Numerical Pain Rating Scale; SE-CSSMS, Self-Efficacy of Cancer Symptom Self-Management Scale; SGCQ, Shortened General Comfort Questionnaire; VAS-A, Visual Analogue Scale for Anxiety.

control groups will receive envelopes containing random numbers generated using the randomisation procedure. The outcome assessor (the nurse appointed by the researcher to be responsible for outcome data collection) and data analysts will be blinded to the participant allocation. Due to the technical nature of the VR interventions, it was difficult to blind the study participants. Thus, to minimise potential methodological bias, the outcome assessor and data analysts will be blinded.[41]

### Patient recruitment
Recruitment will be conducted from April to July 2023. Patients who are scheduled for chemotherapy will be assessed by the nurse on duty, who will review their latest laboratory results, medical records, vital signs and ECG results, to determine whether they are eligible to undergo chemotherapy. The patients who pass the screening process will be asked whether they are willing to be involved in the study and will be asked to sign the informed consent form. Patients who are willing and who sign the informed consent form will be given a sealed envelope containing a random number indicating their group allocation. The participants will be evaluated (pretest) by the outcome assessor (research nurse) before they enter the chemotherapy room. After the evaluation, the researcher will open each participant's envelope and see the random number that they have been assigned (without the knowledge of the participants or the outcome assessor). If the number assigned is between 1 and 40, the participant will be directed to the chemotherapy room on the east side of the hospital (intervention group); if the number is between 41 and 80, the participant will be directed to

the chemotherapy room on the west side of the hospital (control group). We will for the participants in the intervention and control groups to be directed to different chemotherapy rooms, and a research assistant (RA) will accompany the participants to the rooms to minimise contamination of information between participants, between participants and intervention providers and between the nurses or doctors on duty.[42]

### Intervention (S-VR plus standard care)
Participants in the intervention arm will receive immersive VR intervention with a VR box (Shinecon 6.0 VR Box Virtual Reality Glasses with headphones). S-VR is a virtual relaxation and distraction therapy administered through a smartphone-based VR device (head-mounted display) mounted to the head of the patient with cancer during chemotherapy. The intervention involves a virtual environment in which 360° videos of natural panoramas with traditional and classical music (non-copyright) are provided. The S-VR device is paired with a smartphone with either an Android (minimum generation 11, Android V.4.4 KitKat) or iOS (V.12, minimum screen size of 5.5 inch and maximum of 6.0-inch, screen resolution of at least 1080×1920 pixels) operating system and the ability to connect to Wi-Fi or the internet and with the YouTube app installed, which they will use to access the 360° videos. The S-VR content comprises original 360° videos approximately 10 min long and produced by the researchers by using video editing software. The video duration was determined on the basis of a previous study by Fabi *et al*[43] in which this duration was demonstrated to be effective in preventing patient saturation related to content and preventing motion sickness. The 360° videos present panoramic views of Yogyakarta, Indonesia, and Taipei, Taiwan (figure 2). The RA will provide information about the S-VR intervention for approximately 5 min, assist in attaching the S-VR devices to the patient and conduct the intervention for approximately 10 min.

### Control (leaflet plus standard care)
In the control group, participants will be provided with standard care and guided imagery leaflets, which will provide information on the meaning, benefits and methods of completing guided imagery relaxation therapy during chemotherapy. The RA will guide the participants through the practices of the guided imagery relaxation therapy described in the leaflet for ±10 min (figure 3).

### Compensation
Compensation will be provided by the researchers to all respondents after the chemotherapy session in the form of a package containing food, beverage and a small towel. These objects are placed in a small bag along with a note of appreciation for participation in this research.

### Safety and side effect management
The intervention of this study is not associated with any significant side effects because it does not involve

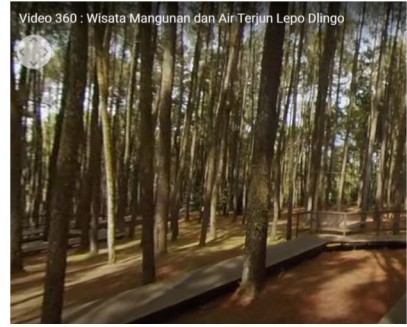
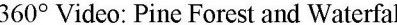
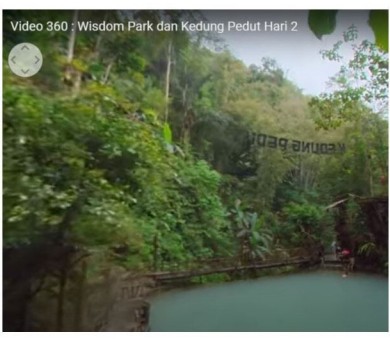
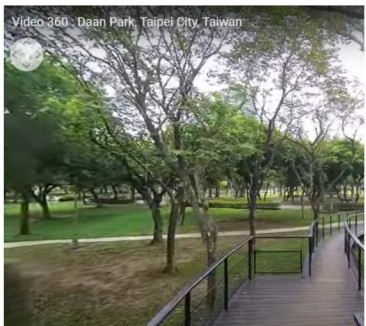

360° Video: Pine Forest and Waterfall in Yogyakarta, Indonesia (https://www.youtube.com/watch?v=OJ5LpBqiRk8)

360° Video: Wisdom Park and Natural Pond in Yogyakarta, Indonesia (https://www.youtube.com/watch?v=6A1RTps5fdY)

360° Video: Daan Park in Taipei City, Taiwan (https://www.youtube.com/watch?v=-h8JgPONKws)

**Figure 2** 360° Videos used in the smartphone-based virtual reality relaxation intervention.

invasive procedures and is administered for a short period (one chemotherapy cycle). However, we predict that the participants will feel discomfort both while using the VR and during chemotherapy and will record their levels of discomfort on a monitoring sheet. The Virtual Reality Symptom Questionnaire, which was developed by Ames et al,[44] will be administered to patients after the VR intervention in order to assess the possibility of cybersickness, a form of motion sickness caused by exposure to VR. The questionnaire evaluates eight general physical side effects (general discomfort, fatigue, boredom, drowsiness, headache, dizziness, concentration difficulties and nausea) and five visual effects (tired eyes, aching eyes, eyestrain, blurred vision and difficulties focusing). The scoring was modified to a 4-point scale (the original score consisted of a 6-point scale), from 0 to 4 (represented of none, mild, moderate, severe) with 0 indicating the absence of the symptom and 4 indicating its severity. During the study, the researchers will clearly indicate that they can be contacted should the participants have any VR-related problems or complaints or if something unexpected occurs.

**Data analysis**

The collected data will be processed using IBM SPSS statistical software V.23 for Windows through editing, coding and data entry processes. The statistician will be blinded to the study outcomes and will conduct analyses on the basis of the labelling of the groups. For the treatment group, the demographic and clinical variables recorded at enrolment will be tabulated. The descriptive statistics will be presented as numbers, means, SDs, medians, minimums and maximums. Homogeneity and normality tests will be conducted before the bivariate analysis test. A homogeneity test for categorical data (age, sex, education level, marital status, occupation, income, the presence of a family member or caregiver during chemotherapy, type of cancer, stage of cancer, duration of cancer diagnosis, chemotherapy cycle and ECOG Scores) will be performed using the $\chi^2$ test or Fisher's test, whereas a homogeneity test for numerical data (height and weight) will be performed using Levene's test if the data are normally distributed and using the Mann-Whitney U test if the data are not normally distributed. Both groups will be considered to be homogeneous if $p > 0.05$. Bivariate analysis will be used to determine the differences in the mean scores for comfort level and symptom management self-efficacy before and after treatment in the intervention and control groups. If the data distribution is normal, we will use a paired t-test and independent t-test. However, if the data distribution is not normal, the non-parametric Wilcoxon and Mann-Whitney U tests will be used if significant differences are identified ($p < 0.05$). For anxiety, pain and vital-sign (blood pressure and pulse rate) outcome measurement analysis, General Linear Model-Repeated Measure (GLM-RM) statistics will be used. The GLM-RM is a statistical test used for the comparative analysis of two unpaired numerical data groups with more than two measurements and is applied when the effect of a treatment or intervention is measured at different time points. If the data are not normally distributed, the repeated Mann-Whitney alternative test with Bonferroni correction will be used. The present study will use statistical method to adjust for potentially confounding effects of the S-VR intervention on the patient's outcomes. These confounders include age, gender, education, cancer stage, chemotherapy cycle, assistance during chemotherapy and comorbidities. Multivariate model can handle identified confounders and minimise the

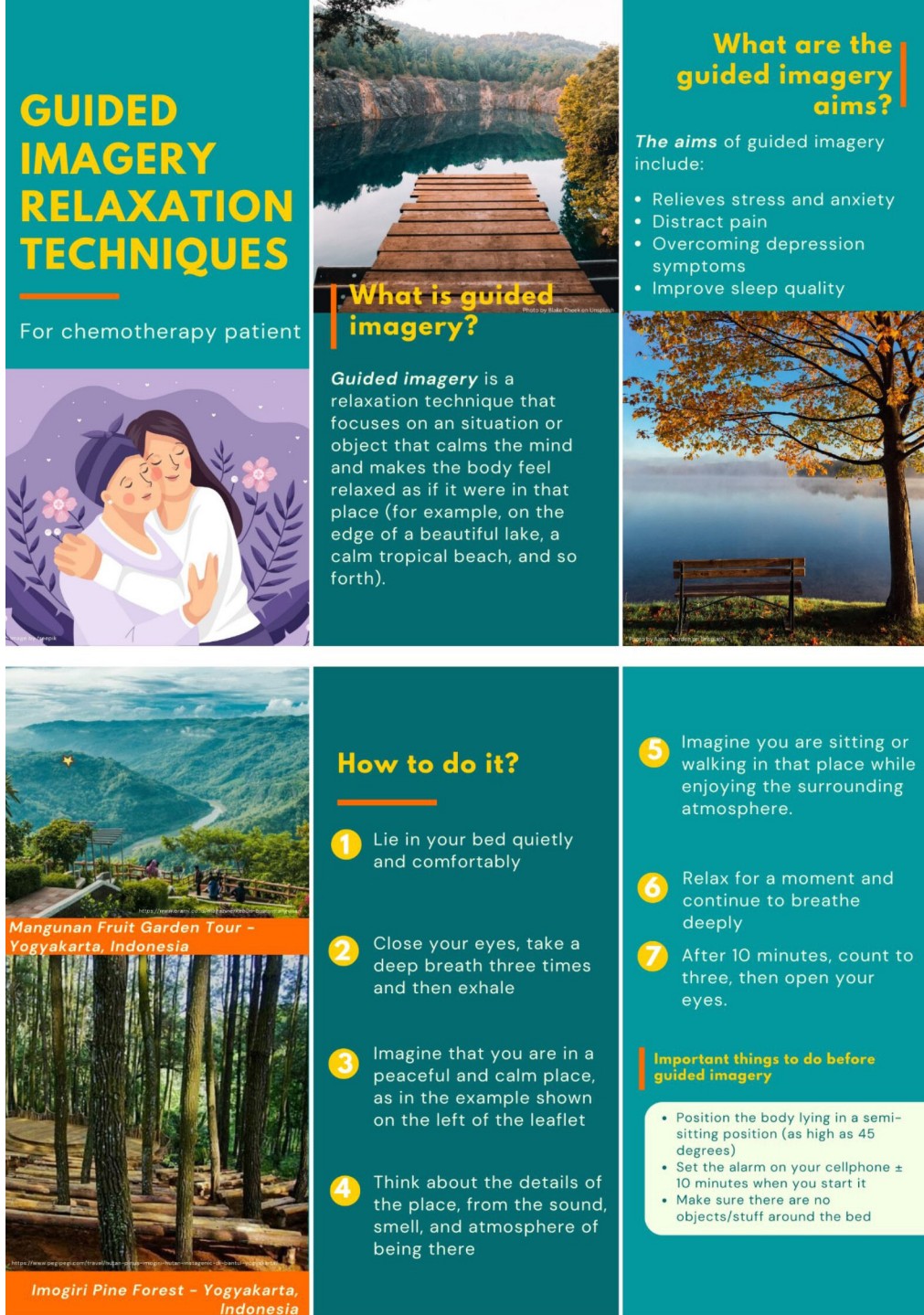

**Figure 3** Guided imagery leaflet.

bias effect of the intervention.[45] Linear regression analysis will be applied by using the enter method. Adjustment by using this analysis can clarify that how much the confounders in the model distort the intervention and outcome.[46]

### Ethics and dissemination

The present study was approved by the Medical and Health Research Ethics Committee of the Faculty of Medicine, Public Health and Nursing Universitas Gadjah Mada—Dr. Sardjito General Hospital, Yogyakarta, Institutional Review Board (approval number: KE/FK/0301/EC/2023). All participants enrolled in the study will be asked to provide written informed consent, which will include a statement requesting their willingness to allow the data to be shared. All information related to the identity of the participants will be kept confidential by the researchers. No post-trial care has been planned. The study results will be

published in a peer-reviewed scientific journal. The results will not indicate the identity of any participant. Authorship eligibility was based on an individual making substantial contributions to the study and manuscript review. Access to the final data set will be granted to the primary investigator and coinvestigators. The study protocol is available on clinicaltrials.gov (ClinicalTrials.gov Identifier: NCT05756465). The study is scheduled to start recruitment at Dr. Sardjito General Hospital in Indonesia in April 2023. After the implementation of the intervention, data collection and statistical analysis, we expect to submit the results to a peer-reviewed journal in October 2023.

**Author affiliations**
[1]School of Nursing, College of Nursing, Taipei Medical University, Taipei, Taiwan
[2]Master of Nursing Program, Faculty of Medicine Public Health and Nursing, Universitas Gadjah Mada, Yogyakarta, Daerah Istimewa Yogyakarta, Indonesia
[3]Department of Surgical Medical Nursing, Faculty of Medicine, Public Health and Nursing, Universitas Gadjah Mada, Yogyakarta, Daerah Istimewa Yogyakarta, Indonesia
[4]Research Center in Nursing Clinical Practice, Department of Nursing, Taipei Municipal Wan-Fang Hospital, Taipei, Taiwan
[5]Department of Nursing, Wan Fang Hospital, Taipei Medical University, Taipei, Taiwan
[6]Cochrane Taiwan, Taipei Medical University, Taipei, Taiwan

**Acknowledgements** This manuscript was edited by Wallace Academic Editing.

**Contributors** MSNG, HH and T-WH conceptualised and designed the study. T-WH and HH provided methodological advice related to the study design. The manuscript was written by MSNG and T-WH and reviewed by all authors. All authors designed the statistical analysis. All authors made critical revisions to the manuscript and provided final approval for the manuscript.

**Funding** This work was supported by the National Science and Technology Council in Taiwan grant number NSTC 111-2622-E-038-003 for APC support.

**Competing interests** None declared.

**Patient and public involvement** Patients and/or the public were involved in the design, or conduct, or reporting, or dissemination plans of this research. Refer to the Methods section for further details.

**Patient consent for publication** Not applicable.

**Provenance and peer review** Not commissioned; internally peer reviewed.

**ORCID iD**
Tsai-Wei Huang http://orcid.org/0000-0002-6722-1153

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
