## [Reviewer comments · BMJ Open]

ARTICLE DETAILS

TITLE (PROVISIONAL)	Efficacy of Smartphone-Based Virtual Reality Relaxation in Providing Comfort to Cancer Patients Undergoing Chemotherapy in Oncology Outpatient Setting in Indonesia: Protocol for a Randomized Controlled Trial
AUTHORS	Huang, Tsai-Wei; Gautama, Made Satya Nugraha; Haryani, Haryani

VERSION 1 – REVIEW

REVIEWER	Tanriverdi, Muberra Bezmialem Vakif Universitesi
REVIEW RETURNED	11-May-2023

GENERAL COMMENTS	This is a protocol to Efficacy of Smartphone-Based Virtual Reality Relaxation in Providing Comfort to Patients with Cancer Undergoing Chemotherapy: Protocol for a Randomized Controlled Trial. The scientific impact of the study is weakened by the insufficient background information, a lack of justification for the significance of conducting such a study, and numerous methodological issues.
---

REVIEWER	King, Amanda L. National Cancer Institute
REVIEW RETURNED	24-May-2023

GENERAL COMMENTS	General comments to authors ▪ Thank you for the opportunity to review this interesting paper. The concept of VR use to manage symptoms in cancer patients is novel and needed in this population, so I was excited to see a randomized controlled trial with this kind of aim. That being said, there are some methodological questions (listed in detail below) that I think need to be thought about and addressed to ensure validity of findings and to provide better clarity about your study design. I had a few questions in the intro as well (also listed below). With some revision, I think this paper would contribute to the literature and is overall well-written. Intro 1. How exactly does VR overcome some of the common limitations of traditional therapy? Can you explain what those limitations are?2. The way you present use of smartphone-based VR In
---

	the intro, it seems like there is already evidence that this kind of approach will improve comfort in those with chemotherapy, so what is your study adding to the literature? Methods  1. How did they come up with your control group condition (guided imagery relaxation therapy via a leaflet plus standard care) ... shouldn't there have been music playing for the control group so the only difference between them is the immersive quality of using VR? 2. Your primary outcome is comfort, but the way you defined it sounds like pain and anxiety are somewhat encompassed in this concept too, so it seems like you are double dipping a bit with your outcome measures. 3. Why are you not getting user feedback on the intervention? That's a big part of assessing not only the acceptability of tech-based interventions, but also about potential side effects the patients can experience when using VR. 4. What degree of visual or cognitive impairment is exclusionary and who determines this (clinician vs. patient report)? And why is wearing glasses a problem (can't you still wear them while watching the VR)? 5. Why does a history of drug addiction problematic for study eligibility? This needs to be explained because it has nothing to do with the intervention. 6. Why are you measuring BP, HR and anxiety at 3 additional timepoints compared to self-efficacy and comfort? 7. What is the compensation amount for participation? This should be disclosed to assess the risk of coercion and bias in results. 8. There are plenty of potential side effects that have been reported in past VR studies, so I don't think it's appropriate to say there aren't any, this should be measured in some way so they can be reported in the results paper. 9. What confounders are you planning to control for in multivariate models?
--	---

VERSION 1 – AUTHOR RESPONSE

Reviewers' comments:

1. Please revise the title of your manuscript to include the setting. This is the preferred format of the journal.

Response 1:

Thank you for your comment. We have added information about the research setting to the title. The

title is "Efficacy of Smartphone-Based Virtual Reality Relaxation in Providing Comfort to Patients with Cancer Undergoing Chemotherapy in the Oncology Outpatient Setting: Protocol for a Randomized Controlled Trial".

2. Please revise the 'Strengths and limitations of this study' section of your manuscript (after the abstract). This section should contain up to five short bullet points, no longer than one sentence each, that relate specifically to the methods. The novelty, aims, results or expected impact of the study should not be summarised here.

Response 2:

Thank you for your suggestion. We revised our strengths and limitations as follows:

"Strengths and Limitations of this Study

- In addition to patient-reported outcomes, the present randomized controlled trial will assess patients' bio-parameters, such as blood pressure and heart rate, as well as cybersickness symptom evaluation due to the VR device.
- We will increase the sample size and make it larger than minimum sample size to increase statistical power.
- Using multivariate statistical analysis, potential confounding variables that may influence the primary outcome will be adjusted for.
- Due to the inherent properties of VR intervention, blinding is inapplicable.
- The principal objective of this study was to enhance patient comfort during the brief duration of chemotherapy within the chemotherapy room."

3. Please include the planned start and end dates for the study in the methods section.

Response 3:

Thank you for the advice. We revised the Method section as follows:

" Recruitment will be conducted from April to July 2023."

4. Along with your revised manuscript, please include a copy of the SPIRIT checklist indicating the page/line numbers of your manuscript where the relevant information can be found (<http://www.spirit-statement.org/>)

Response 4:

Agreed. We have attached the SPIRIT checklist and adapted it to the protocol. Please refer to the attachment.

BMJ Peer Review Comments/Questions for Authors

General comments to authors

Comment: Thank you for the opportunity to review this interesting paper. The concept of VR use to manage symptoms in cancer patients is novel and needed in this population, so I was excited to see a randomized controlled trial with this kind of aim. That being said, there are some methodological questions (listed in detail below) that I think need to be thought about and addressed to ensure validity of findings and to provide better clarity about your study design. I had a few questions in the intro as well (also listed below). With some revision, I think this paper would contribute to the literature and is overall well-written.

Response:

Thank you for taking the time to review our manuscript and for your positive feedback on the novelty and importance of our study on the use of VR to manage symptoms in cancer patients. We appreciate your detailed comments and questions regarding the methodology and introduction of our paper.

We will carefully consider and address each of your points to ensure the validity of our findings and to provide better clarity about our study design. We are committed to revising our manuscript to make it a valuable contribution to the literature.

Thank you again for your constructive feedback.

Intro

Comment 1: How exactly does VR overcome some of the common limitations of traditional therapy? Can you explain what those limitations are?

Response 1: Thank you for your suggestion. We revised the Introduction section to more clarify the statement as follows:

“Virtual Reality (VR) therapy is gaining popularity in alternative medicine due to its ability to overcome certain limitations of traditional therapy. Firstly, VR offers an immersive and interactive environment that fosters a sense of presence and transports individuals to a different reality, potentially reducing anxiety and pain perception (Rubio-Tamayo, et al., 2017). Research has shown that this immersive experience can create a therapeutic effect. Secondly, VR therapy provides a customizable and controlled environment that can be tailored to meet the specific needs of individual patients (Boeldt, et al., 2019). This personalized approach allows for a more targeted and effective treatment experience. By adapting the virtual scenarios and activities to the patient's preferences and requirements, VR therapy offers a unique and individualized intervention. Although VR has been partially applied in cancer care, especially to reduce patient anxiety and pain during invasive procedures.”

Comment 2: The way you present use of smartphone-based VR In the intro, it seems like there is already evidence that this kind of approach will improve comfort in those with chemotherapy, so what is your study adding to the literature?

Response 2:

Thank you for your comment. While previous studies have suggested that smartphone-based VR may improve comfort in patients undergoing chemotherapy, our study adds to the literature in several ways.

First, while previous studies have implied that VR could improve comfort, they did not use comfort as a primary outcome. Our study specifically measures the impact of VR on comfort as one of our primary outcomes. Second, our study builds on previous research conducted in Indonesia (Jadmiko et al., 2022), which found positive results using a similar outcome measure. However, our study uses a more rigorous randomized controlled trial design, compared to the quasi-experimental design used in the Indonesian study.

Based on a preliminary study through a literature review, research using VR interventions, especially in health services in Indonesia, is still limited. Study by Yadi et al. (2019) conducted at a hospital in Lampung reported that VR as a distraction therapy effectively reduces pain in post-laparotomy patients. A case study conducted at a hospital in Yogyakarta by Gautama et al. (2021) presented VR with 360-degree panoramic videos and relaxing music audio. The patient showed a positive perception and decreased patient anxiety. The VR applied in this study uses a smartphone as a medium that displays 3D visuals. However, the results obtained are not effective and reliable enough due to the small sample size and low level of evidence.

Finally, our study proposes several improvements to the VR intervention model and methodology, using a more robust approach and including additional study outcomes. We believe that these factors make our study a valuable contribution to the literature on the use of smartphone-based VR to improve comfort in patients undergoing chemotherapy.

Methods

Comment 1: How did they come up with your control group condition (guided imagery relaxation therapy via a leaflet plus standard care) ... shouldn't there have been music playing for the control group so the only difference between them is the immersive quality of using VR?

Response 1:

Thank you for your question. We chose guided imagery relaxation therapy via a leaflet plus standard care as the control group condition because it is a commonly used and well-established intervention for managing anxiety and stress in patients undergoing chemotherapy.(Charalambous et al., 2015; Roffe et al., 2005)

We did not include music in the control group because we wanted to isolate the effect of the immersive quality of using VR, as you mentioned. By not including music in the control group, we can more accurately assess the impact of VR compared to standard care alone.

We believe that this approach provides a fair and rigorous comparison between the VR intervention group and control group, allowing us to accurately assess the effectiveness of VR in improving comfort and other outcomes in patients undergoing chemotherapy.

Comment 2: Your primary outcome is comfort, but the way you defined it sounds like pain and anxiety are somewhat encompassed in this concept too, so it seems like you are double dipping a bit with your outcome measures.

Response 2:

Thank you for your comment. According our operational definition in our study, we define comfort as a lack of pain, distress, worry, and uneasiness. We measure comfort in four dimensions - physical, social, psychospiritual, and environmental - using a questionnaire. This allows us to assess the overall level of comfort experienced by cancer patients undergoing chemotherapy. While pain and anxiety are related to comfort and are encompassed within our definition of the concept, we also measure these outcomes separately using specific questionnaires. Pain is measured as the unpleasant sensory and emotional sensations experienced by cancer patients during one cycle of chemotherapy due to cancer diagnosis, invasive procedures, and acute side effects of cytotoxic agents. Anxiety is measured as the transitory emotional reactions of cancer patients that arise in situations during one cycle of chemotherapy that are perceived as threatening.

By measuring pain and anxiety separately from comfort, we can assess the impact of our intervention on each of these outcomes individually. This allows us to determine whether our intervention is effective in reducing pain and anxiety specifically, as well as improving overall comfort. We believe that this approach provides a more comprehensive understanding of the impact of our intervention on the comfort of cancer patients undergoing chemotherapy.

Comment 3:Why are you not getting user feedback on the intervention? That's a big part of assessing not only the acceptability of tech-based interventions, but also about potential side effects the patients can experience when using VR.

Response 3:

Thank you for your comment. We agree that user feedback is an important part of assessing the acceptability and potential side effects of tech-based interventions such as VR.

In our study, we have added an assessment of potential side effects when using VR. After the VR intervention, patients will complete the Virtual Reality Symptom Questionnaire (VRSQ), which was developed by Ames in 2005. This questionnaire evaluates eight general physical side effects and five visual effects, allowing us to assess the possibility of cybersickness, a form of motion sickness caused by exposure to VR. This assessment is described in the Safety and Side Effect Management section of our manuscript.

In addition, we have conducted a pilot study to examine the feasibility, acceptability, and preliminary efficacy of our VR intervention. Based on the results of this pilot study, we have made some improvements to our intervention protocol. The pilot study found that our intervention is acceptable, feasible, and safe.

We believe that these measures provide a rigorous and comprehensive assessment of the acceptability and potential side effects of using VR in our study population.

"we predict that the participants will feel discomfort both while using the VR and during ... The Virtual

Reality Symptom Questionnaire (VRSQ), which was developed by Ames in 2005,... During the study, the researchers will clearly indicate that they can be contacted should the participants have any problems or complaints or if something unexpected occurs.”

Comment 4: What degree of visual or cognitive impairment is exclusionary and who determines this (clinician vs. patient report)? And why is wearing glasses a problem (can't you still wear them while watching the VR)?

Response 4:

Thank you for your question. In our study, the degree of visual or cognitive impairment that is exclusionary is determined by a nurse researcher (outcome assessor), who assesses the patient's ability to listen, read, and understand our instructions and information. Assessment of the patient's understanding and their use of spoken and body language forms an important part of the mental state examination. It is helpful to know whether communication behavior and use of language have recently changed. (Boardman et al., 2014)

Therefore, in our study, if a patient has difficulty with these tasks, we assume that they have cognitive impairment and may not be eligible to participate in the study. While this approach is not clinically recommended, we believe that it is sufficient for our purposes.

We removed wearing glasses as one of the exclusion criteria.

Comment 5: Why does a history of drug addiction problematic for study eligibility? This needs to be explained because it has nothing to do with the intervention.

Response 5:

Agreed. We removed the drug or drug addiction as one of the exclusion criteria.

Comment 6: Why are you measuring BP, HR and anxiety at 3 additional timepoints compared to self-efficacy and comfort?

Response 6:

Thank you for your question. In our study, we measure comfort and self-efficacy for symptom management at two time points: at baseline and posttest. We measure anxiety, pain, and vital signs (pulse rate and blood pressure) at five time points: thrice at pretest and twice at posttest.

The reason for measuring these bio-parameters at additional time points is based on previous research that suggests that changes in these parameters are associated with the effectiveness of VR interventions in inducing relaxation. For example, Ioannou et al. (2022) found that changes in blood pressure and heart rate were associated with the effectiveness of VR in inducing relaxation.

Verzwyvelt et al. (2021) also suggested that measuring these bio-parameters is essential as an indicator of the efficacy of VR interventions in stressful situations for cancer patients.

By measuring these bio-parameters at additional time points, we can more accurately assess the impact of our VR intervention on relaxation and stress reduction in patients undergoing chemotherapy.

Comment 7: What is the compensation amount for participation? This should be disclosed to assess the risk of coercion and bias in results.

Response 7:

Thank you for your inquiry. In our study, participants are compensated with a package containing food, beverage, and a small towel. These objects were placed in a small bag along with a note of appreciation for participation. This compensation is provided to participants only after they have completed all research courses, not during the research itself.

After potential participants agree to participate and comprehend the research procedures, we elucidate the compensation. We have disclosed this information in the manuscript's compensation section. We believe that this form of compensation is appropriate, not excessive, and does not pose a risk of coercion or bias in our results.

Comment 8: There are plenty of potential side effects that have been reported in past VR studies, so I don't think it's appropriate to say there aren't any, this should be measured in some way so they can be reported in the results paper.

Response 8:

Thank you for your valuable feedback. We agree that there are potential side effects associated with VR and it is important to measure and report them in our study. In response to your comment, we have revised our protocol to include an assessment of VR side effects. Please refer to the updated section on safety and management of side effects for more information. We appreciate your input in helping us improve our study.

Comment 9: What confounders are you planning to control for in multivariate models?

Response 9:

Thank you for your question. In our multivariate models, we plan to control for several confounding variables that we have identified through a literature review of patients with cancer undergoing chemotherapy. These confounders include age, gender, education, cancer stage, chemotherapy cycle, assistance during chemotherapy, and comorbidities. We believe that controlling for these variables will help us better understand the relationship between our independent and dependent variables and improve the validity of our results.

References

- Boardman, L., Bernal, J., & Hollins, S. (2014). Communicating with people with intellectual disabilities: a guide for general psychiatrists. *Advances in Psychiatric Treatment*, 20(1), 27–36. <https://doi.org/DOI: 10.1192/apt.bp.110.008664>
- Boeldt, D., McMahon, E., McFaul, M., & Greenleaf, W. (2019). Using Virtual Reality Exposure Therapy to Enhance Treatment of Anxiety Disorders: Identifying Areas of Clinical Adoption and Potential Obstacles . In *Frontiers in Psychiatry* (Vol. 10). <https://www.frontiersin.org/articles/10.3389/fpsy.2019.00773>
- Charalambous, A., Giannakopoulou, M., Bozas, E., & Paikousis, L. (2015). A Randomized Controlled Trial for the Effectiveness of Progressive Muscle Relaxation and Guided Imagery as Anxiety Reducing Interventions in Breast and Prostate Cancer Patients Undergoing Chemotherapy. *Evidence-Based Complementary and Alternative Medicine*, 2015, 270876. <https://doi.org/10.1155/2015/270876>
- Jadmiko, A. W., Kristina, T. N., Sujianto, U., Prajoko, Y. W., Dwiantoro, L., & Widodo, A. P. (2022). A Quasi-experimental of a Virtual Reality Content Intervention for Level of Comfort of Indonesian Cancer Patients. *CIN: Computers, Informatics, Nursing*, 40(12). https://journals.lww.com/cinjournal/Fulltext/2022/12000/A_Quasi_experimental_of_a_Virtual_Reality_Content.7.aspx
- Maples-Keller, J. L., Bunnell, B. E., Kim, S.-J., & Rothbaum, B. O. (2017). The Use of Virtual Reality Technology in the Treatment of Anxiety and Other Psychiatric Disorders. *Harvard Review of Psychiatry*, 25(3), 103–113. <https://doi.org/10.1097/HRP.000000000000138>
- Roffe, L., Schmidt, K., & Ernst, E. (2005). A systematic review of guided imagery as an adjuvant cancer therapy. *Psycho-Oncology*, 14(8), 607–617. <https://doi.org/https://doi.org/10.1002/pon.889>
- Tennant, M., Anderson, N., Youssef, G. J., McMillan, L., Thorson, R., Wheeler, G., & McCarthy, M. C. (2021). Effects of immersive virtual reality exposure in preparing pediatric oncology patients for radiation therapy. *Technical Innovations & Patient Support in Radiation Oncology*, 19, 18–25. <https://doi.org/https://doi.org/10.1016/j.tipsro.2021.06.001>
- Verzwyvelt, L., McNamara, A., Xu, X., & Stubbins, R. (2021). Effects of virtual reality v. biophilic environments on pain and distress in oncology patients: a case-crossover pilot study. *Scientific Reports*, 11(1), 20196. <https://doi.org/10.1038/s41598-021-99763-2>

VERSION 2 – REVIEW

REVIEWER	King, Amanda L. National Cancer Institute
REVIEW RETURNED	29-Jun-2023

GENERAL COMMENTS	Thanks for the opportunity to review your manuscript. Use of VR for symptom management is becoming a hot topic and is a really innovative strategy for cancer patients, particularly with situational distress/anxiety during treatment, diagnostic imaging, etc. So this is a timely paper and the trial has sound rationale and design. I would recommend publication with minor revisions. A few things I would consider in your revision. First, I would do a re-read of the paper because in some sentences there seems to be words missing or it is a sentence fragment (an example would be on page 9 in the last sentence of the top paragraph). Second, I think you need to explain the difference between "comfort" and other outcome measures you are assessing that you defined as being directly related to comfort (like anxiety, pain, etc.). It was confusing to me what the difference was between those outcome measures and the reason you separated those variables out to be measured during the chemotherapy infusion. So some clarity as to your rationale would be helpful here. And lastly, I think you could discuss the limitations in greater depth because there are some you aren't mentioning. You aren't assessing the patients' pre-existing psychological conditions, so some patients might be more likely to benefit from VR use for comfort than others. There is also concern for a lack of generalizability given the single institution design, which warrants mentioning.
--

VERSION 2 – AUTHOR RESPONSE

Reviewer: 2

Dr. Amanda L. King, National Cancer Institute

Comments to the Author:

Thanks for the opportunity to review your manuscript. Use of VR for symptom management is becoming a hot topic and is a really innovative strategy for cancer patients, particularly with situational distress/anxiety during treatment, diagnostic imaging, etc. So this is a timely paper and the trial has sound rationale and design. I would recommend publication with minor revisions.

General response:

Dear Dr. Amanda,

Thank you for taking the time to review our manuscript and for your positive feedback. We appreciate your recognition of the timeliness and innovation of our research on the use of VR for symptom management in cancer patients. We are glad to hear that you find our trial to have a sound rationale and design. We will carefully consider your suggestion for minor revisions and make the necessary changes before publication. Thank you again for your valuable input.

Comments 1:

A few things I would consider in your revision.

1. First, I would do a re-read of the paper because in some sentences there seems to be words missing or it is a sentence fragment (an example would be on page 9 in the last sentence of the top paragraph).

Response 1:

Thank you for your careful review of our manuscript and for bringing this issue to our attention. We have thoroughly re-read the paper, particularly the sentence on page 9 that you mentioned, and found that there are no missing words. However, we understand that the sentence structure may have caused confusion and we revised it for clarity. We appreciate your valuable feedback and will continue to improve the manuscript.

Comment 2:

2. Second, I think you need to explain the difference between "comfort" and other outcome measures you are assessing that you defined as being directly related to comfort (like anxiety, pain, etc.). It was confusing to me what the difference was between those outcome measures and the reason you separated those variables out to be measured during the chemotherapy infusion. So some clarity as to your rationale would be helpful here.

Response 2:

Thank you for your comment. We appreciate your feedback and suggestions for improvement. We agree that the difference between "comfort" and other outcome measures related to comfort (such as anxiety, pain, etc.) needs to be clarified in the article. Here is our revised explanation:

Comfort is the primary outcome of this study, and it is a key concept in the context of Chatarina Kolcaba's middle range theory, namely "comfort theory," which is defined as a noun or adjective, and the result of quality care focusing on the patient or family (Smith & Parker, 2015). Comfort is defined as a state that gives rise to satisfaction and relief as a result of being independent of uncomfortable situations and conditions (e.g., anxiety, pain, grief, suffering, etc.) initiated by agents (usually nurses) through the identification and elimination of sources of discomfort before or during the impact on an individual patient or family (Boudiab & Kolcaba, 2015). According to the literature, Kolcaba (2003) and Alligood (2014) identified three types of comfort: relief (a state in which a patient's needs are met), ease (a state of calm and contentment), and transcendence (a state in which one can overcome problems or pain).

According to Kolcaba (2003), there are four comfort contexts: physical, sociocultural, psychospiritual, and environmental. Physical/physiological comfort refers to whether comfort is related to physical sensations or sensations and body functions. Sociocultural comfort refers to patient comfort related to the meaning of interpersonal, family, and social relationships (Alligood, 2014). Psychospiritual comfort refers to self-esteem, self-concept, sexuality, meaning of life, and relationship with a higher power/creator. Environmental comfort refers to the external surroundings that affect the patient's comfort.

In contrast, anxiety and pain are secondary outcomes of this study, and they are specific indicators of the patient's experience of discomfort or distress in different domains. For example, anxiety is a feeling of fear, nervousness, or worry that may affect the patient's mental and emotional well-being. Pain is a sensation of physical suffering or discomfort caused by illness, injury, or treatment. These outcome measures are not synonymous with comfort, but rather reflect some of the factors that may influence the patient's comfort level. Therefore, we separated these variables out to be measured during the chemotherapy infusion to assess how they may affect the patient's comfort in different contexts.

We hope this explanation clarifies the difference between "comfort" and other outcome measures related to comfort in my article. Thank you for your time and attention.

Comment 3:

3. And lastly, I think you could discuss the limitations in greater depth because there are some you aren't mentioning. You aren't assessing the patients' pre-existing psychological conditions, so some patients might be more likely to benefit from VR use for comfort than others. There is also concern for a lack of generalizability given the single institution design, which warrants mentioning.

Response 3:

Thank you for your comment. We appreciate your feedback and suggestions for improvement. We

agree that the limitations section could be expanded to include more potential sources of bias and error. Here is my revised discussion of the limitations:

One of the limitations of this study is that it will not assess the patients' pre-existing psychological conditions, such as depression, anxiety, or post-traumatic stress disorder, which may affect their response to Virtual Reality (VR) intervention. Some patients may be more likely to benefit from VR use for comfort than others, depending on their psychological state and coping skills. Future studies should consider screening the patients for psychological conditions and adjusting the analysis accordingly.

Another limitation of this study is that it will conduct in a single institution, which may limit the generalizability of the findings to other settings and populations. The characteristics of the patients, the staff, the environment, and the VR equipment may vary across different institutions, and may influence the effectiveness and feasibility of VR intervention for comfort. Future studies should replicate the study in multiple institutions with diverse samples to enhance the external validity and applicability of the results.

We hope this discussion addresses your concerns and improves the quality of my article. We revised our manuscript accordingly to include these limitations. Thank you for your time and attention.